# Sleep Disorders in Cancer—A Systematic Review

**DOI:** 10.3390/ijerph182111696

**Published:** 2021-11-07

**Authors:** Antje Büttner-Teleagă, Youn-Tae Kim, Tiziana Osel, Kneginja Richter

**Affiliations:** 1Institute of Cognitive Science, Woosuk University Samnye-up, Samnye-up, Wanju-gun 565-701, Jeollabuk-do, Korea; 2Department of Psychiatry, University Witten-Herdecke Witten, 58455 Witten, Germany; 3Department of Early Childhood Education, Woosuk University Samnye-up, Samnye-up, Wanju-gun 565-701, Jeollabuk-do, Korea; motologie3@gmail.com; 4School of Psychology and Clinical Languages, University of Reading, Reading RG6 6UR, UK; tizianaosel@gmail.com; 5Outpatient Clinic for Sleep Disorders and Tinnitus, University Clinic for Psychiatry and Psychotherapy, Paracelsus Medical University, 90419 Nuremberg, Germany; 6Faculty for Social Work, Technical University Nuremberg Georg Simon Ohm, 90489 Nuremberg, Germany; 7Faculty for medical sciences, University Goce Delcev Stip, 2000 Stip, North Macedonia

**Keywords:** sleep, sleep disorders, sleep disturbances, insomnia, sleep-related breathing disorder (SRBD)/obstructive sleep apnea syndrome (OSAS), narcolepsy and restless legs syndrome (RLS), REM-sleep behavior disorder (REM-SBD), cancer, fatigue

## Abstract

Introduction: Sleep disorders, especially insomnia, are very common in different kinds of cancers, but their prevalence and incidence are not well-known. Disturbed sleep in cancer is caused by different reasons and usually appears as a comorbid disorder to different somatic and psychiatric diagnoses, psychological disturbances and treatment methods. There can be many different predictors for sleep disturbances in these vulnerable groups, such as pre-existing sleep disorders, caused by the mental status in cancer or as side effect of the cancer treatment. Methods: A systematic literature review of 8073 studies was conducted on the topic of sleep and sleep disorders in cancer patients. The articles were identified though PubMed, PsycInfo and Web of Knowledge, and a total number of 89 publications were qualified for analysis. Results: The identified eighty-nine studies were analyzed on the topic of sleep and sleep disorders in cancer, twenty-six studies on sleep and fatigue in cancer and sixty-one studies on the topic of sleep disorders in cancer. The prevalence of sleep disturbences and/or sleep disorders in cancer was up to 95%. Discussion: Sleep disturbances and sleep disorders (such as insomnia, OSAS, narcolepsy and RLS; REM-SBD) in cancer patients can be associated with different conditions. Side effects of cancer treatment and cancer-related psychological dysfunctions can be instigated by sleep disturbances and sleep disorders in these patients, especially insomnia and OSAS are common. An evidence-based treatment is necessary for concomitant mental and/or physical states.

## 1. Introduction

Sleep disturbances and different sleep disorders (e.g., insomnia and sleep-related breathing disorder (SRBD)/obstructive sleep apnea syndrome (OSAS)) are common and considerable complaints of cancer patients. Narcolepsy, restless legs syndrome (RLS) and REM-sleep behavior disorder (REM-SBD) are rarely found. Up to 95% of cancer patients complain of sleep disturbances/disorders during diagnosis, treatment and after 10 years of survivorship. Sleep disturbances/disorders and excessive daytime sleepiness (EDS) have been reported to influence fatigue [1,2,3,4,5,6,7,8,9,10,11,12,13,14,15,16,17,18,19,20,21,22,23,24,25,26,27] and its perceptions. Savard et al. studied cancer survivors and showed that 52% of them reported sleeping difficulties, and 58% reported that cancer either caused or aggravated their sleeping problems [28,29,30,31,32,33,34,35,36,37,38,39,40,41,42,43,44,45,46,47,48,49,50,51,52,53,54,55,56,57,58,59,60,61,62,63,64,65,66,67,68,69,70,71,72,73,74,75,76,77,78,79,80,81,82,83,84,85,86,87,88], especially [58].

Disturbed sleep appears before, while and after cancer diseases. The personalized treatment of the most frequent sleep disorders, e.g., insomnia or sleep-related breathing disorder, could improve both their mental and physical health, specifically for diseases such as cancer. The analyses for this review were very challenging, specifically with regards to systematizing the complex and nonhomogeneous literature about sleep, sleep disturbances and different sleep disorders, their prevalence and the severity of sleep complaints in cancer patients, especially because the cancer population is very heterogenous.

The aim of this systematic review was to evaluate critically the prevalence, severity and efficacy of treatments in cancer-related sleep disorders (CRSD).

## 2. State-of-the-Art

### 2.1. Sleep Disturbances in the Case of Cancer-Related Fatigue (CrF) 

In spite of severe cancer-related fatigue (CrF) [1,2,3,4,5,6,7,8,9,10,11,12,13,14,15,16,17,18,19,20,21,22,23,24,25,26,27] and its perceptions [43,54,58,63,64,67] in cancer patients, there is often also a high prevalence of sleep disturbances (30–50%) in which the proportion of poor sleep or bad sleep quality is significantly higher than in the general population [6,21,23,58,64] (Table 2). Due to frequent “naps” during the day caused by CrF, an additional increase in nocturnal problems can observed [1].

For the research of sleep and quality of sleep, the easy-to-use actigraphy is commonly used [89,90]. Actigraphy data from various studies have shown that there is a strong correlation between the changes in subjectively experienced CrF and sleep quality [2,10,16]. Therefore, CrF-induced sleep disorders can be used as a well-quantifiable CrF-induced event to diagnose and control the course of CrF. Table 1 shows the four sleep-specific phenotypes according to which patients with chronic fatigue syndrome can be classified by means of the more elaborate, but more informative, polysomnography [11].

### 2.2. Insomnia in Cancer 

Insomnia is a very common and frequent comorbidity in cancer patients. The cancer-related insomnia rate is nearly three times higher than that in the general population. Different analyses have shown that 30–50% (up to 95%) of cancer patients have severe sleep difficulties, such as insomnia symptoms or insomnia syndromes (Tables 3–5). Cancer-related insomnia is characterized by a delayed sleep onset, sleep maintenance disorders, reduced total sleep time and/or early-morning awakenings and is associated with excessive daytime sleepiness, fatigue, impaired performance and daytime wellbeing. Furthermore, we established a connection between insomnia and pain, depression, anxiety and/or a reduced quality of life [27,43,53,54,58,63,64,65]. Various types of treatments for insomnia include pharmacological therapies (e.g., hypnotica, sedativa, antidrepressiva, neuroleptics, antihistamine, hormones (melatonin) and herbal extracts) [28,30,42,44,48,57] and nonpharmacological therapies (like Psychoeducational intervention, Cognitive Behavior Therapy (CBT), Professionally administered CBT (PCBT), Video-based CBT (VCBT), Behavioral Therapy (BT), Individualized Sleep Promotion Plan (ISPP), Mindfulness-Based Stress Reduction (MBSR), Valencia model of Waking hypnosis, Internet intervention/Sleep Healthy Using The internet (SHUTi), Progressive Muscle Relaxation (PMR), Autogenic Training (AT), (Electro)Acupuncture (EA), Tai Chi Chih (TCC), Cool Pad Pillow Topper (CPPT), Combined multimodal-aerobic Treatment (CT), Multimodal Treatment (MT) and Aerobic Treatment (AeT)) [29,31,32,33,34,35,36,37,38,39,40,41,44,46,47,49,50,51,52,55,56,57,59,61,62,66,67,68,69,70,71]. Most of the patients with comorbid cancer-related insomnia (that means around 25–50%) are treated pharmacologically [31]. Especially, cancer patients have many side effects and sevaral physical problems from this kind of treatment, so there are numerous limitations that emerge from these pharmacological treatments. Such side effects generally include headaches, dizziness, fatigue, excessive daytime sleepiness and residual daytime sedation and could be potentiated in cancer patients [31]. There is a need and use of complementary and alternative medical methods in cancer patients with cancer-related insomnia. Recent research has shown that complementary and alternative treatments may provide a clinically relevant benefit in cancer-related insomnia [29,31,32,33,34,35,36,37,38,39,40,41,44,46,47,49,50,51,52,55,56,57,59,61,62,66,67,68,69,70,71].

### 2.3. Sleep-Related Breathing Disorder (SRBD)/Obstructive Sleep Apnea Syndrome (OSAS) in Cancer 

Sleep-related breathing disorders (SRBD), especially obstructive sleep apnea syndrome, (OSAS) are common disorders that are characterised by repetitive interruptions of ventilation during sleep. They are caused by recurrent (upper) airway collapses and follwed by sleep fragmentation, intermitted hypoxia and oxidative stress. Systemic and vascular inflammations with endothelial dysfunctions cause diverse multiorgan chronic morbidities and mortalities that affect the cerebrovascular, cardiovascular and metabolic systems in the progress to cancer. Sleep-related breathing disorders are an independent risk factor for cerebrovascular diseases, cardiovascular diseases, metabolic diseases and cognitive decline and are associated with high rates of morbidity and mortality [72,73,74,75,76,77,78,79,80,81,82].

Chronic and intermittent hypoxias seem to play a key role in the regulation of various stages of tumor formation and their progressions. In recent years, some important studies have shown that OSAS patients tend to have a higher prevalence and incidence of cancer and even a higher prevalence of cancer-related mortality [72,73,74,75,76,77,78,79,80,81,82]. One article was able to show that early CPAP treatment can reduce these prevalences: In vitro studies have shown that, in OSAS, there are pro-oncogenic hypoxia properties that are mediate mainly by enhanced posttranslational HIF effects. Intermittant hypoxia results in the increased expression of vascular endothelial growth factor (VEGF) and in tumor growth and metastasis. An effective OSAS treatment coud prevent cancer, its growth and/or metastasis [74] (Tables 3 and 6).

### 2.4. Narcolepsy in Cancer 

The cancer risk as a comorbidity profile of narcoleptic patients has been rarely analyzed [83,84,85] (Tables 3 and 7). There exist only two case studies, and one evaluated the Taiwan nationwide database. Tseng et al. researched the risk of cancer (incidence) among adult narcoleptics [85]. They found that adult narcoleptic patients have a higher risk for developing cancer, but the study was not able to describe the underlying mechanisms for this [83,84,85]. Further research is needed to understand the association between narcolepsy and the development of cancer.

### 2.5. Restless Legs Syndrome (RLS) in Cancer 

Decreased sleep quality, sleep disturbences and/or sleep disruption are very common in cancer patients, especially when they receive chemotherapy [1,2,3,4,5,6,7,8,9,10,11,12,13,14,15,16,17,18,19,20,21,22,23,24,25,26] (Tables 3 and 7). Until now the processes and their pathophysiology have not been completely understood, but most likely, they are multifactorial [86]. Additionally, disturbed sleep and sleep disorders like insomnia and OSAS as disorders and/or diseases with pain, fatigue and mood disturbances often occur in clusters. These clusters can negatively impact the quality of life and the outcome of diseases [1,2,3,4,5,6,7,8,9,10,11,12,13,14,15,16,17,18,19,20,21,22,23,24,25,26]. Sleep disturbance, fatigue and mood disorders (like depression and anxiety) can be based on distinct biologic processes. These processes could be the trigger for inflammatory signaling as a contributing factor of restless legs syndrome (RLS) [86].

The prevalence and/or incidence of restless legs syndrome in cancer is insufficiently researched. A recent study of Saini et al. showed that RLS is frequent in patients with cancer during chemotherapy. They demonstrated that the prevalence is approximately double compared to the normal population (around 18%). In most cases, restless legs syndrome was correlated with depression, anxiety and a decreased quality of life [86].

### 2.6. REM Sleep Behavior Disorder in Cancer 

Rapid Eye Movement Sleep Behavior Disorders (REM-SBD) and cancer are very seldom reported [83,87,88] (Tables 3 and 9). REM-SBD are forms of parasomnias. They are characterised by severe dream-related behavior and increased abnormal electromyographic activity during REM sleep. Sometimes, they are associated with nightmares and parvor nocturnus [83,87,88]. The excessive electromyographic activity during REM sleep reflects the dysfunction of the brainstem structures in REM-SBD patients [87]; acutely, they can be caused by different medications, such as antidepressants or anticholinergic drugs [88]. 

## 3. Method

### 3.1. Data Sources

This review was guided by the Preferred Reporting Items for Systematic Reviews and Meta-Analyses (PRISMA) reporting process where applicable [91].

A systematic literature search was carried out on January 2019 of the databases PubMed, PsycInfo and Web of Knowledge (Figure 1).

The search terms included the following keywords and keyword combinations (sleep OR sleep quality OR sleep disorders OR insomnia OR sleep-related breathing disorder OR obstructive sleep apnea syndrome OR narcolepsy OR restless legs syndrome) OR REM sleep behavior disorder (REM-SBD) AND (cancer) (AND (fatigue)) in English. The keywords were combined as pairs, e.g., sleep disorders AND cancer.

In addition, the reference lists of all of the obtained studies were evaluated. Hard copies of all of the articles were obtained, and they were fully read.

For the analyses of sleep disorders in cancer, only studies from the period 1999/2000–2018 were included in the review, with three exceptions: two studies about sleep and cancer-related fatigue (CrF) in cancer from 1983 to 1993 and a study about OSAS and cancer from 1988. For the analysis of sleep and fatigue in cancer, we even included some older ones.

The 8073 publications were found in the three databases—498 articles were read, and a total number of 89 publications were included in the final analysis.

### 3.2. Types of Studies

Randomized controlled trials (RCTs) and quasi-randomized controlled trials (qRCTs), prospective and retrospective studies, cross-sectional surveys, uncontrolled studies and controlled trials without randomization methods, a special article and case studies were included in this systematic review, because important literature was very rare and inconsistent. We only excluded any forms of qualitative studies.

### 3.3. Types of Participants

Participants who were diagnosed with a sleep disorder (insomnia, sleep-related breathing disorder (SRBD)/obstructive sleep apnea syndrome (OSAS), narcolepsy, restless legs syndrome (RLS) and REM-sleep behavior disorder (REM-SBD)) due to cancer (regardless of gender and age) were included.

### 3.4. Types of Intervention

The review included studies that evaluated different types of insomnia interventions: nonpharmacological interventions—Psychoeducational intervention, Cognitive Behavior Therapy (CBT), Professionally administered CBT (PCBT), Video-based CBT (VCBT), Behavioral Therapy (BT), Individualized Sleep Promotion Plan (ISPP)), Mindfulness-Based Stress Reduction (MBSR), Valencia model of Waking hypnosis, Internet intervention/Sleep Healthy Using The internet (SHUTi), Progressive Muscle Relaxation (PMR), Autogenic Training (AT), (Electro)Acupuncture (EA), Tai Chi Chih (TCC), Cool Pad Pillow Topper (CPPT), Combined multimodal-aerobic Treatment (CT), Multimodal Treatment (MT) and Aerobic Treatment (AeT) and pharmacological interventions, for example, —melatonin (hormone), mirtazapine (hypnoticum); herbal extracts—valerian.

### 3.5. Types of Outcomes

#### 3.5.1. Primary Outcomes

1.The prevalence and/or the incidence of sleep disturbences and/or sleep disorders in cancer were evaluated firstly by objective measurements—polysomnography (PSG)/gold standard and polygraphy (PG) for OSAS or actigraphy. The important sleep parameters included total sleep time (TST), time in bed (TIB), sleep efficiency (SE), sleep quality (SQ), sleep onset latency (SOL), wake after sleep onset or total waking time (WASO).2.The prevalence and/or the incidence of sleep disturbences and/or sleep disorders in cancer are measured secondly by subjective measurements—by scales or indices for the sleep quality (e.g., the Pittsburgh Sleep Quality Index (PSQI)) or special sleep disorders: insomnia (e.g., Insomnia Severity Index (ISI), Athens Insomnia Scale (ASI)), OSAS (e.g., Berlin questionnaire), Narcolepsy (e.g., Narcolepsy Symptom Questionnaire (NSQ)) or RLS (International Restless Legs Syndrome Study Group rating scale (IRLS)).

#### 3.5.2. Secondary Outcomes

The effectiveness of insomnia treatments are measured with sleep diaries. Generally, they include various subjective approaches or several items for reflecting the subjective assessment of daily night’s sleep, including the total sleep time (TST), time in bed (TIB), sleep efficiency (SE), sleep quality (SQ), satisfaction of sleep onset latency (SOL), wake after sleep onset, total waking time (WASO), number of awakenings and morning woken-up time.

### 3.6. Selection of Studies and Data Extraction

The databases PubMed, PsycInfo and Web of Knowledge were searched and potentially studies screened: After the initial screening with checking the titles and abstracts, all the full-text articles were read. The articles that were included in the review were identified, and the data, according to predefined criteria, were extracted. Information such as samples (e.g., kind and number of participants), interventions (in the case of insomnia), measuring instruments, measuring times, methods, outcomes and results were obtained and documented from each study.

## 4. Results

Twenty-six studies for the topic of sleep and fatigue in cancer and sixty-one studies for the topic of sleep disorders in cancer were analyzed, one for sleep disorders generally, forty-four studies for the topic “Insomnia in Cancer” (eight for the “Prevalence of Insomnia in Cancer” and thrirty-six for the “Treatment of Insomnia in Cancer”), twelve studies for the topic “Sleep-Related Breathing Disorder (SRBD)/Obstructive Sleep Apnea Syndrome (OSAS) in Cancer”, three studies for the topic “Narcolepsy in Cancer”and one study for the topic “Restless Legs Syndrome (RLS) in Cancer” (Table 2, Table 3, Table 4, Table 5, Table 6, Table 7, Table 8 and Table 9).

## 5. Discussion

Sleep disturbances and sleep disorders in cancer patients are very common and have different backgrounds compared with sleep difficulties in normal populations because of the differences in the risk factors, vulnerability and cancer-specific life events.

A personalized treatment of sleep disorders in patients with cancer could improve both their mental and physical health.

The goal of this review was to illuminate approaches that might influence sleep, sleep quality and sleep disorders in cancer patients and treatment possibilities in cancer-related insomnia. However, before treatment trials in different sleep disorders (insomnia, OSAS, narcolepsy, RLS and REM-SBD) can be started, prospective and objective studies are needed to unterstand the baseline levels of sleep, sleep difficulties and circadian rhythm in cancer. Sleep disruption in cancer can be caused by many different reasons, such as stress, mental disorders (like depression and anxiety), pain and treatment side effects.

Bad sleep quality, the degree of sleep disruption and sleep disorders have a very important impact on cancer and can used as predictors. Sleep disruptions and disruptions in the circadian rhythms affecting the sleep quality and the circadian rhythm themselves can result in a variety of psychological and physiological mechanisms, which can foster the developent and persistance of cancer-related fatigue. The role of naps in fatigued cancer patients is unclear; it could be that naps are not helpful to decrease cancer-related fatigue—they could have the opposite effect [17]. In noncancer patients, it is known that daytime naps reduce the nightly sleep quality and total sleep time.

Although the relationship between fatigue, sleep and circadian rhythms in cancer is known, there is a very small quantity of scientific reseach about this topic, and the quality is mostly very poor. The existing literature and research is inhomogeneous, and there are many methodological limitations: the types of studies (e.g., randomized controlled trials, quasi-randomized controlled trials, prospective and retrospective studies, cross-sectional surveys, uncontrolled studies and controlled trials without randomization methods, a special article and case studies); participants (different kinds of cancer patients—e.g., with or without treatment and with different entities); interventions (in the case of insomnia: nonpharmacological and pharmacological interventions); outcomes (objective and/or subjective measurements) are not comparable and the sample sizes are mostly very small.

Davidson et al. found in a big sample size with nearly a thousand patients that the total prevalence scores of RLS were present in nearly half of the researched cancer patients, of overly sleepy and of insomnia in around one-third of the patients, of sleeping more than usual and repetitive leg movements in almost one-fifth of them and of breathing interruptions in approximately ten percent [27].

The causes of decreased sleep quality; chronic sleep difficulties and the different sleep disorders (insomnia, OSAS, narcolepsy, RLS and REM-SBD) are multifaceted, and in recent studies, the attention that was paid this problem was too insufficient. Until now, the pathogenesis of cancer-related sleep disorders and the development such as the progression of cancer based on sleep disorders has been unclear. More research about these topics is needed to understand the nature, duration and severity of the different sleep disorders in cancer or their relationship with it.

The prevention of sleep disorders generally and in cancer patients especially and an early personalized treatment can contribute to reducing cancer-related fatigue and severe mental disorders (like depression and anxiety) and can possibily prevent the development, preservation and/or aggravation of cancer.

### 5.1. Expert Recommendations

Sleep disturbances; disruptions of the circadian rhythms and different sleep disorders (e.g., insomnia and sleep-related breathing disorder (SRBD)/obstructive sleep apnea syndrome (OSAS)) could be predictors of cancer development and treatment success (look above). Due to that, cancer patients should be screened by sleep anamnesis and/or by sleep diaries, including the structured exploration of predisposing and precipitating cancer factors, and should be diagnosed—in the case of any kind of sleep-wake difficulties—by polysomnography.

Screening should explore unrefreshing sleep: prolonged sleep latency, frequent awakening and reduced sleep efficiency; daytime sleepiness and fatigue; loud snoring; inadequate nightly behavior and/or nightmares.

Both screening and/or the diagnosis of sleep disturbances; disruptions of the circadian rhythm and/or sleep disorders, as well as adequate sleep health education (including sleep hygiene, rules for good sleep quality and information about the consequences of unhealthy and/or untreated sleep disorders for mental and physical health) should be implemented to minimize the health risks caused by sleep disorders.

Tailored programs are needed and could be helpful to reduce cancer-related fatigue and/or severe mental disorders (like depression and anxiety) to support the outcome of the treatment of patients with cancer and comorbid sleep disorders.

Currently, sleep–wake solutions in cancer are mostly aimed only by responding to emergency reasons and based on isolated and/or fragmented interventions, e.g., the treatment of insomnia: cognitive behavioral therapy for insomnia, nightmares: rehearsal therapy and SRBD: CPAP adherence.

Peronalized medical services for cancer patients should include integrated coaching or the early treatment of the most common sleep disorders and web-based telehealth programs [92] to reduce the preservation and/or aggravation of cancer an/or serious implications, including increased cerebrovascular, cardiovascular and/or metabolic diseases; excessive daytime sleepiness and/or cancer-related fatigue.

### 5.2. References Classification

Studies on Sleep and Cancer-related Fatigue (CrF) in cancer/Connection between sleep and fatigue in oncological diseases [1,2,3,4,5,6,7,8,9,10,11,12,13,14,15,16,17,18,19,20,21,22,23,24,25,26]; Sleep Disorders (generally) [27]; Insomnia (total) [28,29,30,31,32,33,34,35,36,37,38,39,40,41,42,43,44,45,46,47,48,49,50,51,52,53,54,55,56,57,58,59,60,61,62,63,64,65,66,67,68,69,70,71]; Sleep-Related Breathing Disorder (SRBD)/Obstructive Sleep Apnea Syndrome (OSAS) [72,73,74,75,76,77,78,79,80,81,82]; Narcolepsy [83,84,85]; Restless Legs Syndrome (RLS) [86]; REM Sleep Behaviour Disorder (REM-SBD) [83,87,88]; Others (Devices & Methods) [89,90,91,92].

## 6. Conclusions

Cancer patients can suffer under different sleep disturbances and sleep disorders, and these difficulties can be associated with different mental and/or physical problems. Side effects of cancer treatment and cancer-related psychological dysfunctions can be triggered it. Especially insomnia and OSAS are very common in cancer. Because of it, an evidence-based and tailored treatment is necessary.

## Figures and Tables

**Figure 1 ijerph-18-11696-f001:**
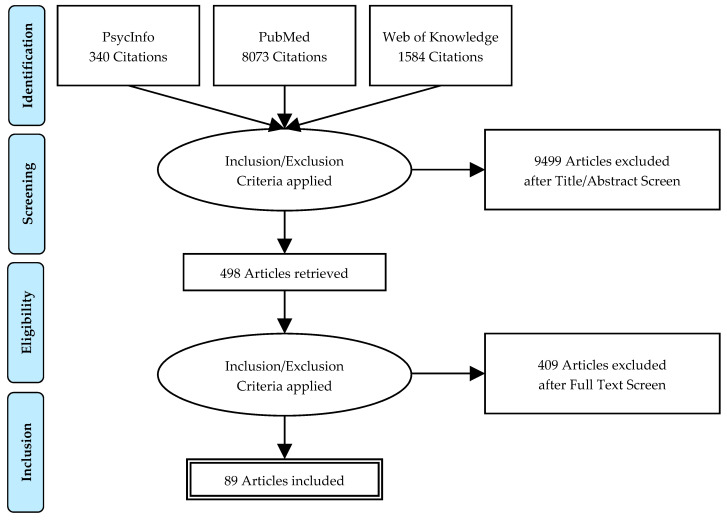
PRISMA flowchart of a systematic review of sleep andand fatigue in cancer.

**Table 1 ijerph-18-11696-t001:** Sleep-specific phenotypes of fatigue (according to Reference [11]).

**First Phenotype**	longer Sleep*Onset*Latency longer REM latencies lower percentage of stage 2 and REM
**Second Phenotype**	more frequent arousals per hour
**Third Phenotype**	longer Total Sleep Time (TST) shorter REM latencies higher percentage of REM and lower percentage of waking time
**Fourth Phenotype**	shortest Total Sleep Time (TST) highest percentage of waking time

**Table 2 ijerph-18-11696-t002:** Studies on Sleep and Cancer-related Fatigue (CrF) in cancer/*Connection between sleep and fatigue in oncological diseases*.

Author	Sample	Fatigue-Measurement	Sleep-/Rhythm-Measurement	Results
Ancoli-Israel et al., *Eur J Cancer Care* 2001; 10 (4): 245–255 [1]	Metaanalysis of existing research: Sleep & Fatigue			
Ancoli-Israel et al., *Support* *Care Cancer* 2006; 14 (3): 201–209 [2]	85 Mamma CA/stages I–IIIA	Multidimen. Fatigue Symptom Inventory (MFSI-SF)	Actigraphy; Sleep diary PSQI; FOSQ	relationship: CrF & subjective sleep (PSQI) no relationship: CrF & objective sleep (Actigraphy)
Ancoli-Israel et al., *Support Care Cancer* 2014; 22: 2535–2545 [3]	68 Mamma CA/stages I–III 60 Controls Data collection at three times: (1) baseline/before chemo (2) end of cycle 4 (3) 1 year post-chemo	Multidimen. Fatigue Symptom Inventory (MFSI-SF)	Actigraphy; PSQI	RCT: Mamma (BC) vs. Controls (NC) BC (all *p* values: <0.05) ○ longer naptime ○ worse sleep quality ○ more fatigue ○ more depressive symptoms ○ more disrupted circadian activity rhythms ○ worse QoL
Banthia et al., *Psychol* *Health* 2009; 24 (8): 965–980 [4]	70 Mamma CA/stages I-IV	5 dimensions CrF (general, physical, mental, emotional fatigue + vigor) Multidimen. Fatigue Symptom Inventory (MFSI-SF)	PSQI; CES-D	PSQI global score signif. correlated with ○ CES-D total score ○ all MFSI-SF subscales (except mental fatigue) PSQI: ↑ SE, ↑ TST, ↓ SOL & ↓ WASO
Berger & Farr, *Oncol Nurs Forum* 1999; 26 (10): 1663–1671 [5]	72 Mamma CA/stages I-II	Piper Fatigue Scale	Actigraphy	disrupted sleep & nightly restlessness during chemotherapy ↑ fatigue/CrF ○ less daytime activity ○ more daytime sleep ○ night awakenings
Chang et al., *Cancer* 2000; 88 (5): 1175–1183 [6]	240 CA	Memorial Symptom Assessment Scale (MSAS)/Item “Lack of Energy”	Functional Assessment Cancer Therapy (FACT-G)/Skala GF/Item Sleep	↑ Fatigue leads to ↑ sleep problems ↑ pain
Cimprich, *Cancer Nurs* 1999; 22 (3): 185–194 [7]	74 Mamma CA	Symptom Distress Scale; POMS	no	higher stress level triad of symptoms ○ insomnia ○ fatigue ○ loss of concentration Insomnia most common complaint (88%) with > 50% by high stress level
Clevenger et al., *Brain* *Behav Immun* 2012; 26 (7): 1037–1044 [8]	136 Ovarian CA – Interleukin-6 (IL-6)	POMS-SF; Multidimen. Fatigue Symptom Inventory (MFSI); Fatigue Symptom Inventory (FSI)	PSQI; Sleep diary	before surgery higher IL-6 → significant relationship (↑ Sleep disorders & ↑ Fatigue) after surgery lower IL-6 → significant relationship (↓ Sleep disorders & ↓ Fatigue)
Engstrom et al., *Cancer Nurs* 1999; 22(2): 143–148 [9]	150 CA – Phase I 42 CA – Phase II	no	Telefon Interview	Phase I: Report—44% poorly sleep Phase II: Report—45% sleep problems (1/2 severe; main problems: nightly awake, ↓ TST, difficulty to fall asleep)
Fiorentino et al., *Drug* *Discov Today Dis Models* 2011; 8 (4): 167–173 [10]	40 Mamma CA/stages I–III	Multidimen. Fatigue Symptom Inventory (MFSI-SF)	Actigraphy	later sleep time & later morning awakening (*rhythm shift*) leads to ↑ Fatigue
Illi et al., *Cytokine* 2012; 58 (3): 437–447 [12]	168 CA Patients 85 Caring relatives – Interleukin-4 (IL-4)	Lee Fatigue Scale	General Sleep Disturbance Scale (GSDS)	phenotype for disease behaviour → role of IL-4 in symptom clusters → 3 classes
Kaye et al., 1983; 114: 107–113 [13]	30 CA28 Cardiological Patients24 Controls	no	Sleep Behaviour Questionnaire	two different chronic diseases → altered sleep patterns → patterns disturbed in different ways CA patients more problems to stay asleep than Controls Cardiac patients more problems difficulty falling asleep, awakened earlier & felt sleepy during day
Liu et al., *Psychooncology* 2009; 18 (2): 187–194 [14]	76 Mamma CA /stages I–III	Multidimen. Fatigue Symptom Inventory (MFSI-SF)	PSQI	significant correlation between Fatigue & Sleep parameters
Liu et al., *Sleep* 2012a; 35 (2): 237–245 [15]	97 Mamma CA/stages I–III	Multidimen. Fatigue Symptom Inventory (MFSI-SF)	Actigraphy; PSQI	comparison T0 & Chemotherapy: → Fatigue ↑ & ↓ SQ → Relationship: + CrF & subjective sleep (PSQI) + CrF & objective sleep (Actigraphy)/TST - CrF & objective sleep (Actigraphy) /Wake daytime
Liu et al., *Brain Behav* *Immun* 2012b; 26 (5): 706–713 [16]	53 Mamma CA/stages I–III – Interleukin-6 (IL-6) – Interleukin-1 Receptor Antagonist (IL-1RA) – C-Reactive Protein (CRP)	Multidimen. Fatigue Symptom Inventory (MFSI-SF)	PSQI	comparison T0 & Chemotherapie: **1.** Fatigue ↑ & ↓ SQ **2.** IL-6 ↑ & ↓ IL-1RA significant Relationship (+) between: Changes **1.** MFSI-SF & IL-6 **2.** PSQI & IL-6 + IL-1RA **3.** WASO & CRP → Tumor-related Fatigue & Sleep Disdorders underlie biochemical mechanism
Miaskowski & Lee, *Journal of Pain and Symptom Management* 1999; 17 (5): 320–332 [17]	24 Bone metastases patients	Lee Fatigue Scale	Actigraphy	Fatigue: ↑ at evening & ↓ at morning; Sleep: ↓ SE; Fatigue associated with - greater inactivity; - fragmented sleep
Mormont et al., *Pathol Biol* 1996; 44(3): 165–171 [18]	30 Colorectal CA	no	Actigraphy	< difference in rest/activity between day & night
Mormont et al., *Clin Cancer Res* 2000; 6 (8): 3038–3045 [19]	200 Colorectal CA	no	Actigraphy	2-years-survivors 5x higher than those with changes in activity rhythms
Morrow, G.R. et al., ???, 1999 (look: at Roscoe et al., *Support Care Cancer* 2002; 10: 329–336) [20]	78 Mamma CA	Multidimentional Assessment of Fatigue; Fatigue Symptom Checklist; POMS	Actigraphy	robust & consistent Circadian rhythms associated with ↓ Fatigue (even after depression)
Mustian et al., *Oncol* *Hematol Rev* 2012; 8 (2): 81–88 [21]	Overview Prevalence: i.e., Fatigue & Sleep			
Owen et al., *Oncol Nurs* *Forum* 1999; 26 (10): 1649–1651 [22]	15 CA	no	Self-Report; PSQI	CA Patients significant ↓ SQ, ↓ SE & ↑ SOL
Palesh et al., *J Clin Oncol* 2009; 28 (2): 292–298 [23]	823 CA (after Chemo)	POMS/Fatigue Inactivity; POMS/Energy; Fatigue Symptom Checklist; Multidimen. Assessment of Fatigue	Hamilton Depression Inventory (HDI)	patients with Insomnia signifcant more symptoms (Depression & Fatigue) compared with patients without Insomnia; differences in entities
Reyes-Gibby et al., *Lancet Oncol* 2008; 9 (8): 777–785 [24]	Overview: Cytokines as markers for Cancer-Related Symptoms	Memorial Symptom Assessment Scale (MSAS)/Item “Lack of Energy”	Functional Assessment Cancer Therapy (FACT-G)/Scale GF/ Item Sleep	Polymorphism in different Cytokine Genes = Potential markers for genetic susceptibility - both cancer risk - as well as cancer symptoms → subgroups for treatment improvement
Roscoe et al., *The Oncologist* 2007; 12 (suppl 1): 35–42 [25]	Review: Cancer-Related Fatigue and Sleep Disorders			
Silberfarb *et al*., *J. Clin* *Oncol* 1993; 11 (5): 997- 1004 [26]	15 Mamma CA 17 Lung CA 32 Insomnics 32 Controls	no	PSG (SE, SOL, WASO)	Lung CA ↓ SE, ↑ SOL & ↑ WASO compared with Mamma CA & Controls

Notes: ???: unclear; ↑: increase; ↓: decrease.

**Table 3 ijerph-18-11696-t003:** Study on Sleep Disorders in cancer.

Author	Sample	Measuring Instrument	Measuring Time	Results
Davidson et al., *Social* *Science & Medicine* 2002; 54: 1309–1321 [27]	982 Cancer patients 303 Breast 108 Gastrointestinal (GI) 155 Genitourinary (GU) 180 Gynecologic (GYN) 114 Lung 123 Skin	Sleep Survey Questionnaire ❖ presence or absence of various sleep phenomena over previous four weeks ❖ questions about - mood - general health - cancer - demographic characteristics	3 months cross-sectional survey study	analyses of sleep disorders pevalence ❖ all cancer ❖ six different cancer types (exept RLS significant differences) ❖ total pevalence score: ❖ 44.3% overly fatigued (lowest: Skin—31.7%, highest: LU—56.1%) ❖ 28.0% overly sleepy (lowest: SKI—18.7%, highest: LU—39.5%) ❖ 18.3% sleeping more than usual (lowest: BR—13.6%, highest: LU—34.2%) ❖ 40.8% RLS (lowest: SKI—35.8%, highest: LU—46.8%) ❖ 16.5% Repetitive Leg Movements (lowest: GYN—12.8%, highest: LU—28.1%) ❖ 30.5% Insomnia (lowest: GU—18.1%, highest: BR—37.8%) ❖ 11.1% Breathing interruptions (lowest: GI—7.4%, highest: SKI—18.7%) ❖ 21.5% use of hypnotica (lowest: SKI—14.6%, highest: LU—40.4%)
			Methods: (1) prevalence of reported sleep problems/six clinics (2) sleep problem prevalence in relation to cancer treatment (3) nature of insomnia (type, duration & associated factors)	

**Table 4 ijerph-18-11696-t004:** Studies on Insomnia in cancer.

Author	Sample	Measuring Instrument(s)	Measuring Time(s)	Results
Graci, *J Support Oncol* 2005; 3 (5): 349–359 [43]	Review: Pathogenesis & Management of Cancer-Related Insomnia			
Howell et al., *Annals* *of Oncology* 2014; 25: 791–800 [45]	Review: grey literature data sources and empirical databases from 2004 to 2012			Review includes: ❖ Practice Guidelines for - sleep–wake disturbances - evaluation and management of chronic insomnia - interventions of sleep disturbances ❖ Randomized Controlled Trials (RCTs) ❖ Cognitive Behavioural Therapy Interventions ❖ Exercise therapy interventions (yoga, walking, home-based exercise)
Minton & Stone, *BMJ S&P Care* 2012; 2: 231–238 [53]	114 Mamma CA - 69 Controls - 45 CRFS	Actigraphy; Insomnia Severity Index (ISI)	between 3 months and 2 years after cancer therapy	Insomnia prevalence significant > in CRFS < in Controls (effect ISI > Actigraphy !)
Park et al., *Sleep Med Res* 2016; 7(2): 48–54 [54]	1248,914 patients analyzed 33,262 were diagnosed with cancer	ICD-10	1-year cross-sectional study	Insomnia was prevalent in 8.21%: 15.2% lung cancer 9.2% non-Hodgkin’s lymphoma 8.8% bladder cancer 8.6% colorectal cancer 8.0% stomach cancer 7.8% prostate, breast & cervix cancer 6.6% liver cancer 5.8% thyroid cancer
Savard et al., *Sleep* 2001; 24 (5): 583–590 [58]	300 Mamma CA	Insomnia Interview Schedule (IIS)—Revised	one time	19% Insomnia syndrome 95% chronic 33% onset of insomnia followed by breast cancer diagnosis 58% cancer either caused or aggravated the sleep difficulties factors associated with an increased risk for insomnia were: ❖ sick leave ❖ unemployment ❖ widowhood ❖ lumpectomy ❖ chemotherapy ❖ a less severe cancer stage at diagnosis
Savard et al., *J Clin* *Oncol* 2009; 27: 5233–5239 [63]	991 CA 466 Mamma 269 Prostata 118 Gynecological	Self-Report Scales; Insomnia Diagnostic Interview	T1—Baseline T2—2 months Tx—6, 10, 14 & 18 months	total: 59.5% 28.5% Insomnia 31.0% Insomnia symptoms; Mamma & Gynecological > Prostata; Insomnia ↓ Therapy course
Savard et al., *J Clin* *Oncol* 2011; 29: 3580–3586 [64]	856 CA 426 Mamma 235 Prostata 96 Gynecological	Insomnia Interview Schedule (IIS)	T1—Baseline Tx—2, 6, 10 & 14 months T6—18 months	total: 59% 28% Insomnia 31% Insomnia symptoms; Mamma & Gynecological > Prostata; Insomnia ↓ Therapy course
Savard & Savard, *Sleep Med Clin* 2013; 8: 373–387 [67]	Review: Insomnia – Cancer – Prevalence – Risk factors – Nonpharmacologic treatment			

**Table 5 ijerph-18-11696-t005:** Studies on Insomnia Treatment in cancer.

Author	Sample	Measuring Instrument(s)	Measuring Time(s) & Method(s)	Results
Barton et al., *J Sup-* *port Oncol* 2011; 9 (1) 24–31 [28]	227 (202) Cancer patients 130 Breast 14 Colon 4 Prostate 52 Other	PSQI; FOSQ (Functional Outcomes of Sleep Questionnaire); BFI (Brief Fatigue Inventory); POMS (Profile of Mood States); TNAS (Toxicity Numeric Analogue Scale); CTCAE (Common Terminology Criteria for Adverse Events)	T1—Baseline T2—Follow-up (4 weeks) T3—Follow-up (8 weeks)	❖ Valerian (Valerian vs. Placebo) - ↓ trouble with sleep - ↓ drowsiness - ↓ fatigue ❖ not differences in: - SQ - toxicities
			Methods: - RCT, dopple-blind - 450 mg Valerian (Herbal Medicine versus Placebo)	
Berger et al., *Psycho-oncology* 2009; 18 (6): 634–646 [29]	219 Cancer patients/ stages I–III	Actigraphy; PSQI; Sleep Diary (SOL, WASO, TIB, TST, SE)	T1—Baseline T2—Follow-up (within 7 days) T3—Follow-up (30 days)	❖ BT group improves: (CBT vs. Controls) - ↓ Sleep Quality (PSQI) - Sleep Diary ↓ SOL, ↓ WASO & ↑ SE ❖ no difference between BT & C: - fatigue
			Methods: - RCT - BT versus Controls (Behavioural Therapy [Individualized Sleep Promotion Plan (ISPP)])	
Chen et al., *Breast* *Cancer Res Treat* 2014; 145 (2): 381–388 [30]	95 Postmenopausal Breast CA/stages 0–III	PSQI; CES-D (Center for Epidemiologic Studies – Depression Scale); NCCTG (North Central Cancer Treatment Group)	T1—Baseline T2—Follow-up (4 months)	❖ Melatonin (Melatonin vs. Placebo) - ↑ SQ - ↑ daytime functions ❖ not differences in: - depression
			Methods: - RCT, dopple-blind - 3 mg Melatonin versus Placebo	
Choi et al., *Integrative Cancer Therapies* 2017; 16 (2) 135–146 [31]	A Systematic Review of Randomized Clinical Trials: Acupuncture for Managing Cancer-Related Insomnia			
Dupont et al., *Health Psychol* 2014; 33 (2): 155–163 [32]	558 Mamma CA	SF-36 (partly); IES-R (Revised Impact of Event Scale); CES-D (Center for Epidemiologic Studies—Depression Scale); PANAS (Positive and Negative Affect Scale) FSI (Fatigue Symptom Inventory); MOS (Medical Outcomes Study); BCPT (Breast Cancer Prevention Trial)	T1—Baseline T2—Post-Treatment (4 weeks) T3—Follow-up (2 months) T4—Follow-up (6 months) T5—Follow-up (12 months)	❖ intrusive thoughts were associated with/ influenced (baseline → 12-month assessment) - higher levels of all symptoms - trajectory of pain - depressive symptoms - negative affect - physical functioning over time ❖ intrusions were not associated with - trajectory of fatigue - sleep - breast cancer-specific symptoms - mental functioning
			Methods: three types of information: (1) print material (2) print material & peer- modeling videotape (3) print material, videotape, 2 education sessions & information workbook	
Epstein & Dirksen, *Oncology Nursing* *Forum* 2007; 34 (5); 51–59 [33]	81 Mamma CA - 40 Controls - 41 CBT-I	Actigraphy; Sleep Diary (SOL, WASO, TIB, TST, SE) PFS (Piper Fatigue Scale)	T1—Baseline T2—Post-CBT-I (6 weeks) T3—Follow-up (12 weeks)	❖ both groups improved (CBT vs. Controls) - Sleep Diary ↓ SOL, ↓ WASO, ↑ SE, ↑ TST & ↑ SQ - Aktigraphy ↓ SOL, ↓ WASO, ↑ SE & ↑ TST ❖ CBT > Controls
			Methods: ❖ RCT ❖ CBT versus no Treatment	
Espie et al., *J of* *Clinical Oncology* 2008; 26: 4651–4658 [34]	150 CA 87 Mamma 34 Prostate 24 Colorectal 5 Gynecological (110 CBT/50 TAU)	PSQI; ESS; Sleep Diary (SOL, WASO, TST, SE); HADS; FSI (Fatigue Symptom Inventory); CrQoL (Cancer-Related Quality of Life); FACT-G (Functional Assessment of Cancer Therapy Scale – General)	T1—Baseline T2—Post-Treatment T3—Follow-up (6 months)	❖ CBT was associated with - reductions in wakefulness of 55 min. per night - moderate to large effect sizes for 5/7 QOL outcomes - ↑ SE, ↑ TST, ↓ SOL & ↓ WASO ❖ TAU no change
			Methods: ❖ RCT ❖ CBT versus TAU (Treatment As Usual)	
Fiorentino et al., *Nature and Science of Sleep* 2010; 2: 1–8 [35]	21 Mamma CA - 11 IND-CBT-I - 10 Controls	Actigraphy; PSQI; Insomnia Severity Index (ISI); Sleep Diary (SOL, WASO, TIB, TST, SE)	T1—Baseline T2—Post-CBT-I (6 weeks) T3—Follow-up (12 weeks)	❖ CBT-I was associated with - ↓ ISI - Aktigraphy & Sleep Diary ↑ SE, ↑ TST & ↓ WASO
			Methods: ❖ RCT ❖ CBT versus no Treatment	
Fleming (Espie) et al., *Psychooncology* 2014; 23 (6): 679–684 [36]	113 Cancer patients with Insomnia - 73 CBT-I - 40 Controls	PSQI; Sleep Diary (SOL, WASO, TIB, TST, SE) HADS; FSI (Fatigue Symptom Inventory)	T1—Baseline T2—Post-Treatment T3—Follow-up (6 months)	❖ CBT was associated with - ↓ clinical insomnia - ↓ clinical fatigue ❖ CBT & TAU - no changes in anxiety ❖ completely symptom free at post-treatment: - 7 (9.6%) in CBT - 0 (0.0%) in TAU
			Methods: ❖ RCT ❖ CBT versus TAU (Treatment As Usual)	
Garland et al., *Contemporary Clinical Trials* 2011; 32 (5): 747–754 [37]	???	Actigraphy; Sleep Diary (SOL, WASO, TST, SE) ???	T1—Baseline T2—Post-Treatment (2 months) T3—Follow-up (3 months)	❖ high prevalence of distress & sleep distur- bances in cancer population ❖ MBSR should produce sleep effects comparable to CBT-I
			Methods: CBT-I versus MBSR (Mindfulness-Based Stress Reduction)	
Garland et al., *J* *Clin Oncol* 2014; 32: 1–9 [38]	327 screened CA 111 randomly assigned 53 Breast 12 Prostate 11 Blood/lymph 10 Female Genitourinary 9 Head & Neck 7 Colon/GI 7 Lung 2 Skin CBT-I: *n* = 47 MBSR: *n* = 64	Actigraphy; PSQI; Insomnia Severity Index (ISI); Sleep Diary (SOL, WASO, TST, SE)	T1—Baseline T2—Post-Treatment (2 months) T3—Follow-up (5 months)	❖ CBT-I was associated with - ↓ ISI ❖ Aktigraphy - CBT-I: ↑ SE, ↑ TST_-_, ↓ SOL & ↓ WASO - MBSR: ↑ SE, ↑ TST_+_ & ↓ WASO ❖ Sleep Diary CBT-I & MBSR - ↑ SE, ↑ TST, ↓ SOL & ↓ WASO
			Methods: CBT-I versus MBSR (Mindfulness-Based Stress Reduction)	
Garland et al., *Neuropsychiatric Disease and Treatment* 2014; 10: 1113–1124 [39]	Review: Efficency of CBT-I in cancer Inclusion of 4 studies			❖ results for ❖ un-controlled studies (*n* = 4) ❖ controlled studies/RCT (*n* = 8) CBT-I in cancer is associated with → clinically improvements in subjective sleep outcomes improved sleep → Improvement in: ❖ mood disturbance ❖ cancer-related fatigue ❖ overall quality of life
Garland et al., *Explore (N.Y.)* 2015; 11 (6): 445–454 [40]	72 Cancer patients MBCR: *n* = 32 CBT-I: *n* = 40	???	T1—Baseline T2—Post-Treatment (? months) T3—Follow-up (3 months)	❖ CBT-I & MBCR: - ↓ Insomnia severity ❖ CBT-I > MBCR: - ↓ dysfunctional sleep beliefs
			Methods: CBT-I versus MBCR (Mindfulness-Based Cancer Recovery)	
Garland et al., *Contemporary Clinical Trials* 2016; 47: 349- 355 [41]	160 Cancer patients with Insomnia	???	T1—Baseline T2—Mid-Treatment (4 weeks) T3—Post-Treatment (8 weeks) T4—Follow-up (3 months)	???
			Methods: ❖ RCT ❖ CBT versus Aucupuncture	
Garland et al., *Sleep Medicine* 2016; 20: 18–24 [42]	88 Cancer patients with Insomnia	ESS; Sleep Diary (SOL, WASO, TST, SE)	T1—Baseline T2—Post-Treatment (7 weeks) T3—Follow-up (3 months)	❖ CBT-I + A & CBT-I + P: → improvement sleep continuity → no difference in Daytime Sleepiness ❖ PLA: → absence of improvement of SL & WASO → trend to increased TST
			Methods: (RCT) (1) CBT-I + P (CBT-I and Placebo) (2) CBT-I + A (CBT-I and Armodafinil) (3) ARM (Armodafinil alone) (4) PLA (Placebo alone)	
Heckler (Garland) et al., *Supportive Care in Cancer* 2016; 24 (5): 2059–2066 [44]	96 Cancer patients with Insomnia	Insomnia Severity Index (ISI); BFI (Brief Fatigue Inventory); FACIT-Fatigue scale	T1—Baseline T2—Post-Treatment (7 weeks) T3—Follow-up (3 months)	❖ **No sleep results !!!** ❖ CBT-I + A & CBT-I + P: → no difference in Fatigue
			Methods: (RCT) (1) CBT-I + P (CBT-I and Placebo) (2) CBT-I + A (CBT-I and Armodafinil) (3) ARM (Armodafinil alone) (4) PLA (Placebo alone)	
Irwin et al., *JNCIM* 2014; No. 50; 295–301 [46]	90 Mamma CA random subsample (*n* = 48)	Blood samples: - C-Reactive Protein (CRP) - Interleukin-6 (IL-6) - Tumor Necrosis Factor-α (TNF) subsample analyzed by genome-wide transcriptional profiling	T1—Baseline T2—Post-Treatment (3 months)	❖ Sleep disruption → increases in TLR-4-activated production of proinflammatory cytokines ❖ no change in systemic inflammation (CRP) ❖ changes in cellular inflammation (IL-6 & TNF) TCC reduced ❖ cellular inflammatory responses (↓ IL-6 & ↓ TNF) ❖ expression of genes encoding proinflammatory mediators
			Methods: CBT-I versus TCC (Tai Chi Chih)	
Kim M. et al., *BMJ open*, 2017; 7 (8): 1- 10 [47]	45 Cancer patients	Actigraphy; Insomnia Severity Inventory (ISI); PSQI; Sleep Diary (SOL, WASO, TIB, TST, SE) BDSS (Blood Deficiency Scoring System); EA (Electroacupuncture); FACT-F (Functional Assessment of Cancer Therapy- Fatigue); MoCA (Montreal Cognitive Assessment)	T1—Baseline T2—Treatment (3 weeks) T2—Post-Treatment (5 weeks) T2—Post-Treatment (9 weeks)	**Without results !!!** *„The result of this study will be published in peer-reviewed journals or presented at academic conferences.“*
			Methods: (4 weeks) EA versus Sham-EA (Electroaccupuncture) versus TAU (Treatment As Usual)	
Kim S.W. et al., *Psychiatry and Clinical Neurosciences* 2008; 62: 75–83 [48]	45 Cancer patients 25 Lung 5 Breast 6 Gastrointestinal tract 3 Hepatobiliary tract 3 Other malignancy	C-LSEQ (Chonnam National University Hospital- Leeds Sleep Evaluation Questionnaire) SF-36; MADRS (Montgomery-Asberg Depression Rating Scale); EuroQoL (EQ) -5D	T1—Baseline T2—Post-Treatment (4 weeks)	mirtazapine rapidly improved sleep disturbance, nausea, pain and quality of life, as well as depression in cancer patients Sleep ↑: ↑ TST, ↓ SOL, ↓ SQ
			Methods: - prospective, open labeled study - 15–45 mg mirtazapine	
Kröz et al., *BMC* *cancer* 2017; 17 (1), 166: 1–13 [49]	126 Mamma CA	PSQI; CFS-D (Cancer Fatigue Scale)	T1—Baseline T2—Post-Treatment (10 weeks) T3—Follow-up (6 months)	❖ T1: MT & CT > AeT ❖ T2: MT or CT > AeT ❖ MT & CT improve: - sleep (PSQI) - fatigue (CSS-D) ❖ MT/T1: ↓ SOL ❖ CT/T2: ↑ SQ, ↓ SOL & ↑ TST
			Methods: (RCT) (a) MT (Multimodal Treatment) (b) CT (MT + AeT) (Combined Treatment) (c) AeT (Aerobic Training)	
Lengacher et al., *Psychooncology* 24 (4): 424–432 [50]	79 Mamma CA /stages 0-III	OSP (Objective Sleep Parameters): - Actigraphy SSP (Subjective Sleep Parameters): - PSQI; - Sleep diary	T1—Baseline T2—Treatment (6 weeks) T2—Post-Treatment (12 weeks)	❖ positive effect of MBSR(BC) on OSP at 12 weeks on: - sleep efficiency: 78.2% MBSR (BC) vs. 74.6% UC, *p* = 0.04 - percent of sleep time: 81.0% MBSR (BC) vs. 77.4% UC, *p* = 0.02 - less number waking bouts: 93.5 MBSR (BC) vs. 118.6 UC, *p* < 0.01 ❖ small positive effects in MBSR(BC) on SSP at 6 weeks - PSQI total score: *p* = 0.09 ❖ no relationship between min. of MBSR (BC) practice and SSP or OSP
			Methods: (RCT) MBSR (BC) vs. UC (Mindfulness-Based Stress Reduction [Breast Cancer]) (Usual Care)	
Marshall-McKenna et al., *Supportive Care in Cancer* 2016; 24 (4): 1821–1829 [51]	74 Mamma CA with Insomnia - 68.9 % pre-menopausal - 31.1% post-menopausal	HADS; FACT-B (Functional Assessment of Cancer Therapy - Breast) sleep/hot flush diaries (over 2-week periods)	T1—Baseline T2—Treatment (x weeks) T3—Post-Treatment (x weeks)	❖ CPPT + SC > SC: - ↓ hot flushes *p* = 0.090 - ↓ HADS depression *p* = 0.036 ❖ no differences in: - FACT-B - HADS anxiety “This study supports the use of the CPPT as an aid to reduce sleep disturbance and the frequen-cy/severity of hot flushes.”
			Methods: (RCT) - Intervention Arm: CPPT + SC (Cool Pad Pillow Topper + Standard Care) vs. - Control Arm: SC (Standard Care)	
Mendoza et al., *Psychooncology* 2017; 26 (11): 1832–1838 [52]	44 Cancer patients	MOOS (Medical Outcomes Survey Sleep); PROMIS (Fatigue) (Problem Index Patient-reported Outcomes Measurement Information System); NRS (Pain intensity) (Numerical Rating Scales)	T1—Baseline T2—Treatment (3 weeks) T3—Post-Treatment (3 months)	VMWH-CBT vs. Controls → beneficial effects of the VMWH-CBT - sleep problems - fatigue - average pain intensity
			Methods: - RCT, cross-over - VMWH-CBT vs. Controls (Valencia model of Waking Hypnosis with Cognitive-Behavioural Therapy)	
Peoples (Garland) et al., *Journal of Cancer Survivorship* 2017; 11 (3): 401–409 [55]	95 Cancer patients with Insomnia	Insomnia Severity Index (ISI); FACT-G (QoL) (Functional Assessment of Cancer Therapy - General)	T1—Baseline T2—Post-Treatment (7 weeks) T3—Follow-up (3 months)	❖ **No sleep results !!!** ❖ CBT-I + A & CBT-I + P: → CBT-I improves QoL → Armodafinil no effect on QoL
			Methods: (RCT) (1) CBT-I + P (CBT-I and Placebo) (2) CBT-I + A (CBT-I and Armodafinil) (3) ARM (Armodafinil alone) (4) PLA (Placebo alone)	
Ritterband et al., *Psychooncology* 2012; 21 (7): 695–705 [56]	28 Cancer patients with Insomnia	Insomnia Severity Index (ISI); Sleep Diary (SOL, WASO, TIB, TST, SE) MFSI-SF (Multidimensional Fatigue Symptom Inventory - Short Form); UQ (Internet Intervention Utility Questionnaire); HADS; SF-12	T1—Baseline T2—Post-Treatment (3 months)	SHUTi vs. Controls → beneficial effects of the SHUTi - ↓ ISI - ↓ HADS - ↑ SF-12 - Sleep Diary: ↑ SE, ↑ TST, ↓ SOL & ↓ WASO (Controls improved a little too: SE & WASO)
			Methods: ❖ RCT ❖ Internet CBT-I/SHUTi (Sleep Healthy Using the Internet) versus WLC (Waiting-List Control)	
Roscoe, J.A., (Garland, Sh.N.) et al., *Journal of Clinical Oncology* 2015; 33 (2): 165–171 [57]	96 Cancer patients with Insomnia	PSQI; Insomnia Severity Index (ISI)	T1—Baseline T2—Post-Treatment (7 weeks) T3—Follow-up (3 months)	CBT-I + A & CBT-I + P: → CBT-I improves Insomnia Severity (ISI) → CBT-I improves Sleep Quality (PSQI) → Armodafinil no effect on Insomnia & SQ
			Methods: (RCT) (1) CBT-I + P (CBT-I and Placebo) (2) CBT-I + A (CBT-I and Armodafinil) (3) ARM (Armodafinil alone) (4) PLA (Placebo alone)	
Savard (Quesnel) et al., *JCCP* 2003; 71 (1): 189–200 [59]	10 Mamma CA	PSG; Insomnia Severity Inventory (ISI); Sleep Diary (SOL, WASO, TIB, TST, SE) MFI (Multidimensional Fatigue Inventory); BDI & STAI; QLQ-C30+ 3 (European Organization for Research & Treatment of Ca. Quality of Life Questionnaire)	T1—Baseline T2—Post-Treatment (3 months) T3—Follow-up (6 months)	CBT was associated with - ↓ ISI: = ↓ Insomnia severity - ↑ PSG & ↑ Sleep Diary: = ↑ SE, ↑ TST, ↓ SOL & ↓ WASO
Savard et al., *Journal* *of Pain and Symptom Management 2004;* 27 (6): 513–522 [60]	24 Mamma CA	PSG; Skin conductance	???	❖ nightly hot flashes - ↑ wake time - ↓ Stage 2 sleep - ↑ REM latency ❖ ↑ sleep disruption ❖ ↑ poor sleep
			Methods: CBT	
Savard et al., *JCO* 2005 I & II; 23 (25): 6083–6096 & 6097- 6106 [61]	57 women with insomnia caused or aggravated by breast cancer	PSG; Insomnia Severity Inventory (ISI); Sleep Diary (SOL, WASO, TIB, TST, SE) MFI (Multidimensional Fatigue Inventory); HADS; QLQ-C30+ 3 (European Organization for Research & Treatment of Ca. Quality of Life Questionnaire); Immune measures: enumeration of blood cell counts (i.e., WBCs, monocytes, lymphocytes, CD3, CD4, CD8, CD16/CD56) & cytokine product. (Interleukin-1-beta [IL-1β], Interferon gamma [IFN-γ])	T0—Pre-Waiting T1—Baseline T2—Post-Treatment T3—Follow-up (3 months) T4—Follow-up (6 months) T5—Follow-up (12 months)	CBT was associated with (post-treatment vs. control patients) - ↓ ISI: = ↓ Insomnia severity - ↑ PSG & ↑ Sleep Diary: = ↑ SE, ↑ TST, ↓ SOL & ↓ WASO - higher secretion and/or level of IFN-γ & IL-1β - lower increase of lymphocytes
			Methods: CBT versus WLC (Waiting-List Control)	
Savard (Tremblay) et al., *JCCP* 2009; 77 (4): 742–750 [62]	57 Mamma CA	PSG; Insomnia Severity Inventory (ISI); Sleep Diary (SOL, WASO, TIB, TST, SE) DBAS (Dysfunctional Beliefs and Attitudes about Sleep Scale); ABS (Adherence to Behavioural Strategies) TEPCQ (Treatment Expectancies and Perceived Credibility Questionnaire); TAPQ (Therapeutic Alliance Perception Questionnaire); HADS	T1—Baseline T2—Post-Treatment (2 months) T3—Follow-up (6 months)	❖ CBT was associated with - ↓ ISI: = ↓ Insomnia severity - ↑ Sleep Diary: = ↑ SE, ↑ TST, ↓ SOL & ↓ WASO ❖ CBT wasn‘t associated with - PSG
			Methods: CBT versus WLC (Waiting-List Control)	
Savard et al., *Psycho-Oncology* 2013; 22 (6): 1381–1388 [65]	60 Prostate CA	Insomnia Severity Index (ISI); PSQ (Physical Symptoms Questionnaire)	T1—Baseline Tx—1, 2, 4, 6, 8 & 12 months T8—16 months	❖ ADT → risk of Insomnia ↑ ❖ side effects of ADT & RTH → development of Insomnia
			Methods: ADT (Androgen Deprivation Therapy) RTH (Radiation therapy)	
Savard (Casault) et al., *Behaviour Research and Therapy* 2013; 67: 45–54 [66]	83 Cancer patients	???	T1—Baseline T2—Post-Treatment T3—Follow-up (3 months) T4—Follow-up (6 months)	❖ CBT was associated with (mCBT vs. Controls) - ↑ all sleep parameters - ↓ dosage of hypnotics - ↓ anxiety & depression - ↓ maladaptive sleep habits - ↓ erroneous beliefs about sleep - ↑ subjective cognitive functioning ❖ therapeutic gains of mCBT-I well sustained up to 6 months
			Methods: mCBT versus no Treatment (minimal CBT)	
Savard et al., *Sleep* 2014; 37 (8): 1305- 1314 [68]	242 Mamma CA	Actigraphy; Insomnia Severity Index (ISI); Sleep Diary (SOL, WASO, TIB, TST, SE)	T1—Baseline T2—Post-Treatment (6 weeks)	❖ PCBT-I & VCBT-I were associated (compared to CTL): - ↑ sleep - ↓ Insomnia severity - ↓ Early Morning Awakenings (EMA) - ↓ depression - ↓ fatigue - ↓ dysfunctional beliefs about sleep ❖ Remission rates of insomnia (ISI < 8) were significantly greater in PCBT-I as compared to VCBT-I: - 71.3% vs. 44.3%, *p* < 0.005
			Methods: (RCT) (1) Professionally administered CBT-I (PCBT-I; *n* = 81) (2) Video-based CBT-I (VCBT-I; *n* = 80) (3) no treatment (CTL; *n* = 81)	
Savard et al., *Sleep* 2016; 39 (4): 813–823 [69]	242 Mamma CA	Insomnia Severity Index (ISI); Insomnia Interview Schedule (IIS); Sleep Diary (SOL, WASO, TIB, TST, SE) MFI (Multidimensional Fatigue Inventory) EORTC QLQ-C30; HADS; DBAS-16 (Dysfunctional Beliefs & Attitudes about Sleep Scale – Abbreviated version);	T1—Baseline T2—Post-Treatment (6 weeks) T3—Follow-up (3 months) T4—Follow-up (6 months) T5—Follow-up (12 months)	❖ PCBT-I > VCBT-I > CTL: - ↑ sleep: ↑ SE, ↑ TST, ↓ SOL & ↓ WASO - ↓ Insomnia severity (ISI, IIS) - ↓ Early Morning Awakenings (EMA) - ↓ depression - ↓ anxiety - ↓ dysfunctional beliefs about sleep - ↑ QoL ❖ remission rates of insomnia (ISI < 8) were significantly greater in PCBT-I as compared to VCBT-I and CTL: e.g., 12 month FU - 67% vs. 59% vs. 48%, *p* < 0.100
			Methods: (RCT) (1) Professionally administered CBT-I (PCBT-I; *n* = 81) (2) Video-based CBT-I (VCBT-I; *n* = 80) (3) no treatment (CTL; *n* = 81)	
Simeit et al., *Suppor-tive Care in Cancer* 2004; 12 (3): 176–183 [70]	229 Cancer patients (breast, kidney or prostate)	???	T1—Baseline T2—Post-Treatment (3-4 weeks) T3—Follow-up (6 months)	❖ PMR & AT vs. CG → improvements over time: - sleep latency (*p* < 0.001) - sleep duration (*p* < 0.001) - sleep efficiency (*p* < 0.001) - sleep quality (*p* < 0.001) - sleep medication (*p* < 0.050) - daytime dysfunction (*p* < 0.050) - quality-of-life ❖ indicate a benefit of rehabilitation treatment in general ❖ no evidence between the two intervention groups
			Methods: (RCT) (1) Progressive Muscle Relaxation (PMR; *n* = 80) (2) Autogenic Training (AT; *n* = 71) (3) Control Group (CG; *n* = 78)	
Zhou et al., *Behavioral Sleep Medicine* 2017; 15 (4): 288–301 [71]	10 (12) Cancer patients	Insomnia Severity Index (ISI); PSQI; Sleep logs [SL]; (SOL, WASO, TIB, TST, SE); SF-12	T1—Baseline T2—Post-Treatment (20 days) T3—Follow-up (2 months after T2)	❖ adapted CBT-I improves: - ↑ Sleep (SL): ↑ SE, ↓ SOL, ↓ WASO & ↓ EMA - ↓ Insomnia severity (ISI) - ↓ Sleep Quality (PSQI) ❖ no effect on: - TST - QoL
			Methods: - Adapted CBT-I 3 x intervention in person (6) and via videoconference (6)	

Notes: ???: unclear; ↑: increase; ↓: decrease.

**Table 6 ijerph-18-11696-t006:** Studies on Sleep-Related Breathing Disorder (SRBD)/Obstructive Sleep Apnea Syndrome (OSAS) in Cancer.

Author	Sample	Measuring Instrument(s)	Measuring Time(s)	Results
Campos-Rodrigues et al., *Am J Respir* *Crit Care Med* 2013; 187 (1): 99–105 [72]	4910 Patients (Multicentric Cohort Study)	PSG/PG	T—Baseline T2—4.5 years	↓ TSat_<90%_ vs. ↑ CA Incidence ↑ AHI vs. ↑ CA Incidence → higher Risk: **1.** < 65 years; **2.** ♂; **3.** no CPAP
Cao et al., *Sleep* *Breath* 2015; 19 (2): 453–457 [73]	Obstructive Sleep Apnea promotes Cancer development and progression (Animal studies)			OSAS = Risk factor ❖ Prevalence of Cancer Cancer-related Mortality Intermittent hypoxia (Sleep fragmentation)? → activation of HIF-1 & VEGF pathways → tumor growth → aggressive cancer behaviour
Dewan et al., *Chest* 2015; 147 (1): 266–274 [74]	Intermittent hypoxemia and OSA: Implications for comorbidities (Animal & Human studies)			Intermittent hypoxemia promotes → Oxidative stress → Inflammation → Increased sympathetic activation → Progression of cancer → **Effect of CPAP !!!**
Faiz et al., *The* *Oncologist* 2014; 19: 1200–1206 [75]	56 Patients with tumors in the head and neck region	PSG	Retrospective review from 2006 to 2011	**1.** SRBD = common in patients with tumors in the head/ neck region → caused by sleep disruption **2.** Architectural changes from tumor and/or therapy lead to OSA
Gomez-Merino et al., *Respiration* 2003; 70: 107–109 [76]	Case study: 55 years old man *non-Hodgkin-Lymphoma*			Symptom development: ❖ Nocturnal apneas ❖ Excessive Daytime Sleepiness (ESS = 18) ❖ Limited daytime activity ❖ AHI = 45.6/h ❖ SaO_2_ = 81% Polysomnogram: ❖ Changes sleep architecture ❖ Sleep Rhythm Disorder ❖ Sleep fragmention (AI = 27.2/h) ❖ nearly no SWS: 0.6% ❖ Sleep efficiency: 82%
Kendzerska et al., *CMAJ* 2014; 186 (13): 985–992 [77]	10,149 Patients	PSG all patients AHI ≥ 5 or suspected OSAS (but AHI < 5)	A) from 1994 to 2010 B) from 1991 to 2013	Methods: ❖ examined association between - Severity of OSA - prevalence and incidence of cancer ❖ controlling for known risk factors for cancer development Result: link between OSA & Cancer development or progression through chronic hypoxemia
Marshall et al., *JCSM* 2014; 10 (4): 355–362 [78]	400 OSAS-Patients	PG/MESAM IV	T1—Baseline 1990 T2—20 years 2010 (Follow-up)	Follow-up: 397 people removed *n* = 4 with a previous stroke from the mortality/ CVD/CHD/stroke analyses (*n* = 393) *n* = 7 with cancer history from the cancer analyses (*n* = 390) 20 years Follow-up ❖ 77 (19.6%) Deaths ❖ 103 (26.2%) Cardiovascular events - 17 fatalities ❖ 31 (7.9%) Strokes ❖ 59 (15.0%) Coronary Heart Diseases ❖ 125 (32.1%) incident of cancer - 39 cancer fatalities **1.** moderate-severe OSA was significantly associated with - all-cause mortality (HR = 4.2; 95% CI: 1.9, 9.2) - cancer mortality (HR = 3.4; 95% CI: 1.1, 10.2) - incident cancer (HR = 2.5; 95% CI: 1.2, 5.0) - stroke (HR = 3.7; 95% CI: 1.2, 11.8) but not significantly with - CVD incidence (HR = 1.9; 95% CI: 0.75, 4.6) - CHD incidence (HR = 1.1; 95% CI: 0.24, 4.6) **2.** mild OSA was associated with a halving in - mortality (HR = 0.5; 95% CI: 0.27, 0.99)
Martinez-Garcia et al., *Eur Respir J* 2012; 40: 1315–1317 [79]	--- Special Article			current insights and perspectives: ❖ apneas / hypopneas ❖ intermittent hypoxias / nocturnal desaturation ❖ sleep fragmention → Cardiovacular diseases → Cerbrovakular diseases → Metabolic diseases → Systemic inflammatory diseases **important role in regulating the various stages of tumor development and progression**
Nieto et al., *Am J Respir Crit Care Med* 2012; 186 (2): 190–194 [80]	1522 Patients (Background: Wisconsin Sleep Cohort Study)	PSG	T1—Baseline T2—22 years	SRBD = associat. with ↑ CA Mortality → higher Risk in ↑ SRBD: **1.** ↑ AHI; **2.** ↓ Tsat_<90%_
Partinen et al., *Chest* 1988: 94 (6): 1200–1204 [81]	198 OSAS Patients (Tracheostomy vs. Weight loss)	PG	Retrospective review from 1972 to 1980	↑ BMI & ↑ AHI → lead to Vascular death ❖ Myocardial infarction ❖ Cerebrovascular accidents **Relationship Cancer & OSA ???** **→ no answer !!!**
Seidell, *Eur J of* *Clinical Nutrition* 2010; 64: 35–41 [82]	Review: Waist circumference and Waist/Hip ratio in relation to all-cause Mortality, Cancer and Sleep Apnea (Human studies)			BMI ↑ → Risk of Cancer ↑ → Risk of OSA ↑ Waist circumference & Waist/Hip ratio → better indicator of all-cause mortality than BMI **Relationship Cancer & OSA ???** **→ no answer !!!**

Notes: ???: unclear; ↑: increase; ↓: decrease; ♂: male.

**Table 7 ijerph-18-11696-t007:** Studies on Narcolepsy in cancer.

Author	Sample	Measuring Instrument(s)	Measuring Time(s)	Results
Adams et al., *Arch Neurol* 2011; 68 (4): 521–524 [83]	Case study: 35 years old man *Testicular cancer*			Symptom development: ❖ Sleep fragmention ❖ Excessive Daytime Sleepiness (ESS = 24) ❖ Cataplexy (?) ❖ Hypnagogic (visual) hallucinations Polysomnogram: ❖ Sleep Rhythm Disorder ❖ Sleep fragmention (AI = 62/h) ❖ no SWS MSLT: 2.2 min. sleep latency with 5 episodes REM*Onset
Landolfi & Nadkarni, *Neuro-Oncology* 2003; 5: 214–216 [84]	Case study: 55 years old man *Tonsil cancer*			Symptom development: ❖ Excessive Daytime Sleepiness ❖ Cataplexy Polysomnogram: ❖ Sleep efficiency: 44.3% ❖ Sleep latency: 4.5 min. ❖ Respiratory Disturbance Index (RDI): 23.1 (hypopneas) MSLT: 9 min. sleep latency with 2 episodes REM*Onset
Tseng et al., *Cancer* *Epidemiol* 2015; 39 (6): 793–797 [85]	2,833 Narcoleptics	???	from 2000 to 2009/National Health Insurance Research Database	adult narcoleptic patients → higher cancer risk (74 Cancer/SIR 1.32; 95% CI, 1.04–1.66, *p* = 0.0248) → ♀ = higher Risk: (SIR 1.52; 95% CI, 1.05–2.13, *p* = 0.026) **1.** ↑ Head & Neck CA (SIR 6.17; 95% CI, 1.66–15.80, *p* = 0.009) **2.** ↑ Gastric CA (SIR 4.87; 95% CI, 1.31–12.48, *p* = 0.020) → underlying mechanism unclear

Notes: ???: unclear; ↓: decrease; ♀: female.

**Table 8 ijerph-18-11696-t008:** Studies on Restless Legs Syndrome (RLS) in cancer.

Author	Sample	Measuring Instrument(s)	Measuring Time(s)	Results
Saini et al., *J Pain Symptom Manage* 2013; 46: 56–64 [86]	173 CA different entities 32.4% Colorectal 17.3% Mamma 7.5% Prostata 6.4% Ovary 5.8% - Bladder - Gastroenteropancreatic Neuroendocrine 3.5 % - Pancreas - Testis - Stomach 2.9% - Lung - Adrenal cortical 2.3% - Uterus - Kidney 1.7% Head & Neck 1.2% Thymus 0.6% - Esophagus - Thyroid	Pittsburgh Sleep Quality Index (PSQI); International Restless Legs Syndrome Study Group rating scale (IRLS); Functional Assessment of Cancer Therapy-General (FACT-G); Hospital Anxiety and Depression Scale (HADS)	T0—before Chemotherapy T1—after Chemotherapy	58.8% Sleep problems (PSQI > 5) 20.0% RLS positive screened RLS & PSQI not associated with Anemia, Neurotoxic chemotherapeutic agents or Benzamides (neuroleptics / antipsychotics) significant correlation PSQI & RLS (*p* = 0.007)

**Table 9 ijerph-18-11696-t009:** Studies on REM Sleep Behaviour Disorder (REM-SBD) in cancer.

Author	Sample	Measuring Instrument(s)	Measuring Time(s)	Results
Adams et al., *Arch* *Neurol* 2011; 68 (4): 521–524 [83]	Case study: 35 years old man *Testicular cancer*	PSG		Polysomnogram: ❖ agitated and vocal behaviour during sleep ❖ sleepwalking ❖ nightly confusion ❖ sleep fragmention (AI = 62/h) ❖ no SWS
Jianhua, Ch. et al., *Intern Med* 2013; 52: 617–621 [87]	Case study: 30 years old man *Brainstem lymphoma* (diffuse large B-cell)	PSG; MRI		Polysomnogram (Sleep Rhythm Disorder): ❖ violent motor and vocal behaviour during sleep ❖ enhanced submental and limb electromyo-graphic tone during REM ❖ increased muscular activity during REM
Shinno, H. et al., *J* *Pain Symptom Manage* 2010; 40 (3): 449–452 [88]	Case study: 3 cases 70–76 years old patients 2 males & 1 femal *Advanced cancer* *(1 x kidney; 2 x stomach*)	PSG		Polysomnogram: ❖ decreased TST ❖ prolonged REM latency ❖ decreased REM ❖ increased muscle activity during REM

## Data Availability

The data that support the findings of this study are available from the corresponding author, upon reasonable request.

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
