# Peer review of "Sleep Disorders in Cancer—A Systematic Review"

_ijerph, 2021, doi:10.3390/ijerph182111696_

Round 1

Reviewer 1 Report

The article titled “Sleep Disorders in Cancer – A Systematic Review” is quite nicely constructed and reads well. Although, it needs to be edited carefully for abbreviations and typos listed below. The paper's synthesis seemed all right except the literature covered is quite outdated (January 2019). In my view, the content of the manuscript presented here seems very strong and scientifically sound. Meanwhile, I am listing out some of the issues/suggestions related to the manuscript to consider.

  1. Line (28), “Insomnia and OSAS should are common” needs to be re-written.
  2. Lines 52-53, These statements should be placed in the fag end of the introduction to summarize the goal of the manuscript.
  3. Many of the abbreviations used in the manuscript e.g., SRBD, OSAS and RLS, are not quite consistent. In general, they are abbreviated consistently many times throughout the article. Authors should be quite consistent with the representation of information.
  4. CPAP in line 114 should be explained or expanded.
  5. REM-RBD is wrong abbreviation in line 139.
  6. Many of the abbreviations like EMG (line 142), qRCTs (line 170) etc., never got used again in the manuscript.
  7. Is there any rational of listing the studies in tables? I personally feel that listing them based on recent to oldest publication order will make more sense. If otherwise, the listing criteria should be mentioned in the method section.

Author Response

Dear reviewers,

thanks for your reading and your good comments. I corrected the language (checked by a native speaker) and I edited the article and marked the most important points.

Best regards

Reviewer 2 Report

This is well-written paper by  Büttner-Teleagă et al. The authors took very interesting topic of sleep disorders in cancer and present it as a systematic review. In my opinion the following issues should be adressed:

- The abstract shouldn't contain the subheadings (introduction, methods, ...
- The citations must be enumerated correctly. It must be started from 1 and go in the order (1,2,3,...); NOT: 27, 43, 54!

- "Varyious types of treatment for insomnia include pharmacological therapies (e.g. Hypnotica, Sedativa, Antidrepressiva, Neuroleptics and Antihistamine; Hormones [Melatonin]; Herbal extracts) (28, 30, 42, 44, 48, 57), and non-pharmacological therapies (like Psychoeducational Intervention, Cognitive Behaviour Therapy (CBT), Professionally administered CBT (PCBT), Video-based CBT (VCBT), Behavioral Therapy (BT) [Individualized Sleep Promotion Plan (ISPP)], Mindfulness-Based Stress Reduction (MBSR), Valencia model of Waking Hypnosis, Internet Intervention / Sleep Healthy Using the Internet (SHUTi), Progressive Muscle Relaxation (PMR)," - TThe capittal letters shouldn't be overused

- line 103: typo: "hyoxias"

- Chapter 1.3. The authors took a very important topic of OSAS. In my opinion, the molecular links between OSAS and cancer should be better described, e.g. in the context of HIF-1 upregulation. As it was shown patients with obstructive sleep apnea present with CHRONIC upregulation of serum HIF-1α protein instead of intermitted hypoxias "caused by recurrent (upper) airway collapses" (see 10.5664/jcsm.8682). This link is especially interesting in the context of the following sentence: "r. Sleep-Related Breathing Disorders are an 106
independent risk factor for cerebrovascular diseases, cardiovascular diseases, metabolic 107
diseases, cognitive decline and are associated with high rates of morbidity and mortality". In the current literature, we can found papers proposing HIF-1a upregulation as a mediator of these comorbidities (10.3389/fphys.2020.01035).

- The citation in tables should be in accordance with the citation style used in the text. Moreover, the meaning of the colors (blue, red) and questions mark (?, ???) must be explained.

- The authors should prepare the section "Strengths and limitation of the study".

Author Response

(The authors gave the same response as above.)

Round 2

Reviewer 1 Report

Following the revisions, I received the revised manuscript for review titled "Sleep Disorders in Cancer – A Systematic Review". I'm glad to see that the authors took reviewers' advice into consideration and worked on manuscript. The current manuscript is quite improved than the one before it and thus recommended for publication!

Reviewer 2 Report

The authors addressed all the comments and sufficiently improved the manuscript.